# Impact of Pancreatic Resection on Survival in Locally Advanced Resectable Gastric Cancer

**DOI:** 10.3390/cancers13061289

**Published:** 2021-03-14

**Authors:** Shih-Chun Chang, Chi-Ming Tang, Puo-Hsien Le, Chia-Jung Kuo, Tsung-Hsing Chen, Shang-Yu Wang, Wen-Chi Chou, Tse-Ching Chen, Ta-Sen Yeh, Jun-Te Hsu

**Affiliations:** 1Department of General Surgery, Chang Gung Memorial Hospital at Linkou, Chang Gung University College of Medicine, Taoyuan 333, Taiwan; b9302071@cgmh.org.tw (S.-C.C.); mp2297@cgmh.org.tw (C.-M.T.); d0100106@cgu.edu.tw (S.-Y.W.); tsy471027@cgmh.org.tw (T.-S.Y.); 2Department of Gastroenterology, Chang Gung Memorial Hospital at Linkou, Chang Gung University College of Medicine, Taoyuan 333, Taiwan; b9005031@cgmh.org.tw (P.-H.L.); m7011@cgmh.org.tw (C.-J.K.); q122583@cgmh.org.tw (T.-H.C.); 3Department of Hematology-Oncology, Chang Gung Memorial Hospital at Linkou, Chang Gung University College of Medicine, Taoyuan 333, Taiwan; f12986@cgmh.org.tw; 4Department of Pathology, Chang Gung Memorial Hospital at Linkou, Chang Gung University College of Medicine, Taoyuan 333, Taiwan; ctc323@cgmh.org.tw

**Keywords:** gastric adenocarcinoma, multiorgan resection, pancreatic resection, survival

## Abstract

**Simple Summary:**

Gastric adenocarcinoma (GC) is the fifth most common malignancy and third leading cause of cancer-related mortality worldwide. Multiorgan resection is necessary to achieve clear R0 margins in GC patients with adjacent organ invasion (T4b). However, whether these patients benefit from aggressive surgery involving pancreatic resection (PR) remains unclear. Here we aimed to evaluate the impact of PR on survival in patients with locally advanced resectable GC. We found that the patients with T4b lesions who underwent PR had poorer survival than those who underwent resection of other adjacent organs. Further pancreaticoduodenectomy did not improve survival in pT3–pT4 GC patients with positive duodenal margins. These findings may be useful to practicing clinicians by aiding optimal decision making for treatment plans and surgical procedures.

**Abstract:**

Whether gastric adenocarcinoma (GC) patients with adjacent organ invasion (T4b) benefit from aggressive surgery involving pancreatic resection (PR) remains unclear. This study aimed to clarify the impact of PR on survival in patients with locally advanced resectable GC. Between 1995 and 2017, patients with locally advanced GC undergoing radical-intent gastrectomy with and without PR were enrolled and stratified into four groups: group 1 (G1), pT4b without pancreatic resection (PR); group 2 (G2), pT4b with PR; group 3 (G3), positive duodenal margins without Whipple’s operation; and group 4 (G4), cT4b with Whipple’s operation. Demographics, clinicopathological features, and outcomes were compared between G1 and G2 and G3 and G4. G2 patients were more likely to have perineural invasion than G1 patients (80.6% vs. 50%, *p* < 0.001). G4 patients had higher lymph node yield (40.8 vs. 31.3, *p* = 0.002), lower nodal status (*p* = 0.029), lower lymph node ratios (0.20 vs. 0.48, *p* < 0.0001) and higher complication rates (45.2% vs. 26.3%, *p* = 0.047) than G3 patients. The 5-year disease-free survival (DFS) and overall survival (OS) rates were significantly longer in G1 than in G2 (28.1% vs. 9.3%, *p* = 0.003; 32% vs. 13%, *p* = 0.004, respectively). The 5-year survival rates did not differ between G4 and G3 (DFS: 14% vs. 14.4%, *p* = 0.384; OS: 12.6% vs. 16.4%, *p* = 0.321, respectively). In conclusion, patients with T4b lesion who underwent PR had poorer survival than those who underwent resection of other adjacent organs. Further Whipple’s operation did not improve survival in pT3–pT4 GC with positive duodenal margins.

## 1. Introduction

Gastric adenocarcinoma (GC) is the fifth most common malignancy, with approximately 950,000 newly diagnosed cases annually and is the third leading cause of cancer-related mortality globally (700,000 deaths per year) [1,2]. Radical resection (R0 resection) with complete eradication of the tumor, macroscopically and microscopically, still remains the cornerstone of therapeutic management of GC. Advances in adjuvant and neoadjuvant chemotherapy, targeted therapy and immunotherapy have also improved outcomes of GC [3,4]. In patients with locally advanced GC involving adjacent organs (T4b) or the distal duodenal bulb, gastrectomy combined with multiorgan resection (MOR), such as pancreaticoduodenectomy (Whipple’s operation), is mandatory to achieve a clear margin, so called R0 resection [5,6,7,8,9]. However, the more of the involved organ are resected, the higher are the rates of surgery-related mortality and morbidity [5,9,10], which may defer patients from receiving further systemic therapy. This situation is especially remarkable in patients undergoing gastrectomy plus pancreatic surgery since pancreatic resection (PR) is associated with considerably high rates of postoperative pancreatic fistula and intra-abdominal infection [6,7,8]. Patient nutrition status or general performance usually deteriorated significantly after Whipple’s operation. Consequently, it has been a matter of debate whether patients with locally advanced resectable disease would benefit from extensive surgery involving MOR.

Although our previous study has shown that patients with T4b lesion who underwent R0 resection had better outcomes than those who underwent R1 or R2 resection, the benefit gains were very limited with 15.1 and 11.1 months of median survival time, respectively [5]. Li et al. reported 26 months of median survival and 59.4% of morbidity rate from a pooled analysis of 69 patients undergoing Whipple’s operation from 13 articles [6]. Furthermore, the Dutch Upper Gastrointestinal Cancer Audit group also published the results on 55 patients undergoing en-bloc PR reporting 15 months of median overall survival (OS) [11]. Nonetheless, little is known concerning whether the impact of PR on outcomes is similar to that of resection of other adjacent organs in locally advanced resectable GC. This study aimed to compare outcomes in locally advanced resectable GC patients undergoing gastrectomy plus PR including Whipple’s operation or distal pancreatectomy with those in patients undergoing gastrectomy and resection of other adjacent organs. Additionally, we aimed to clarify the survival benefit of additional Whipple’s operation to achieve R0 resection in case of positive distal (duodenal) margins.

## 2. Materials and Methods

This study was conducted in accordance with the Declaration of Helsinki (1996), and the protocol was approved by the Ethics Committee (institutional review board) of Chang Gung Memorial Hospital in Taiwan (approval number: 201801640B0C101 and approval date: 17 September 2019).

### 2.1. Inclusion Criteria

We recruited patients with locally advanced (pT3–T4) resectable GC undergoing radical-intent gastrectomy with and without PR between 1995 and 2017 in Chang Gung Memorial Hospital at Linkou. Patients were stratified into four groups according to the PR or distal duodenal margins status: group 1 (G1), pT4b without PR; group 2 (G2), pT4b with PR; group 3 (G3), positive duodenal margins without Whipple’s operation; group 4 (G4), cT4b with Whipple’s operation. The choice between total or subtotal gastrectomy with adjacent organ resection for each patient was decided by the surgeon, based on the tumor location, surgical findings, and resection margin status. The decision on performing Whipple’s operation or not was made based on the surgeon’s judgment or preference after considering the disease severity, patient’s general performance, and surgical risks.

### 2.2. Exclusion Criteria

Patients who died within 30 days after surgery (surgical mortality) or those who died post-surgery during the same hospitalization (hospital mortality) were excluded from the survival analysis.

### 2.3. Data Collection

Clinicopathological data were obtained from a prospectively constructed medical database. Tumors with well or moderate differentiation were defined as differentiated, and those with poor differentiation or signet ring cell histology were defined as non-differentiated. The lymph node ratio (LNR) was calculated by dividing the number of lymph nodes positive for malignancy by the total number of retrieved lymph nodes. The tumors were staged according to the 8th edition American Joint Committee on Cancer staging system [12]. Adjuvant chemotherapy was administered to medically fit patients in 6–8 weeks post-surgery.

### 2.4. Postoperative Follow-Up

Recurrence patterns were categorized as locoregional, peritoneal seeding, or hematogenous/distant, which were defined according to our previously published paper [13]. Postoperative follow-up studies included imaging examinations (such as abdominal computed tomography/ultrasonography, chest plain radiography, and upper digestive endoscopy) and laboratory tests (liver enzymes and tumor markers) every 3 to 6 months in the first 2 years after surgery and every year thereafter.

Cancer-specific disease-free survival (DFS) or OS duration was calculated as the time from surgery to recurrence or death or the date of the last follow-up (30 June 2020). The median follow-up duration was 16.5 months (1.3–260.1).

### 2.5. Statistical Analysis

Data are presented as percentages or the mean ± standard deviation. Clinical records were compared by Student’s *t* test, Pearson’s chi-square test, or Fischer’s exact test as appropriate. Patient DSF or cancer-specific survival rates were calculated using the Kaplan–Meier curve analysis, and the differences between subgroups were assessed using the log-rank test. Factors that were deemed of potential importance in the univariate analysis (*p* < 0.05) were included in the multivariate analysis using Cox proportional hazards model. *p* < 0.05 was regarded as statistically significant. Statistical analyses were performed with SPSS for Windows, version 20 (SPSS, Chicago, IL, USA).

## 3. Results

### 3.1. Demographics and Clinicopathological Features in G1 and G2 Patients

Table 1 demonstrates a comparison of the demographics and clinicopathological features between G1 (*n* = 50) and G2 (*n* = 94) patients. G2 patients had higher rates of perineural invasion than G1 patients (*p* < 0.001). There was no difference in the variables including age, sex, tumor size, tumor location, type of gastrectomy, nodal status, staging, the number of retrieved lymph nodes, LNR, differentiation, and the presence of lymphatic or vascular invasion between two groups. Furthermore, the two groups had similar rates of surgical complications, hospital mortality, patients receiving adjuvant chemotherapy, intervals between surgery to chemotherapy, and chemotherapy cycles.

### 3.2. Predictors of Disease-Free and Overall Survival in G1 and G2 Patients

Univariate analysis indicated that PR, nodal status, stage, LNR, and the presence of lymphatic or perineural invasion were significant prognostic factors for DFS (Table 2) and OS (Table 3) in T4b patients who did or did not undergo PR. In multivariate analysis, G2 patients had a higher risk of recurrence (hazard ratio [HR] = 1.74; 95% confidence interval [CI], 1.116–2.713; *p* = 0.015; Table 2) and cancer-specific death (HR = 1.897; 95% CI, 1.210–2.974; *p* = 0.005; Table 3) than G1 patients. G1 patients had significantly longer median DFS (19.3 vs. 9.3 months, *p* = 0.003) and OS (27.1 vs. 16 months, *p* = 0.004) than G2 patients (Figure 1).

### 3.3. Demographics and Clinicopathological Features in G3 and G4 Patients

Table 4 shows a comparison of the demographics and clinicopathological features between G3 (*n* = 98) and G4 (*n* = 46) patients. G4 patients had more T4b tumors (*p* < 0.001), higher numbers of retrieved nodes (*p* = 0.009), higher rates of complications (*p* = 0.016), lower percentages of N3b and lower LNR (*p* < 0.001) than G3 patients. No difference was identified in age, sex, tumor size, tumor location, type of gastrectomy, staging, differentiation, and lymphatic, vascular, or perineural invasion between the two groups. Percentages of hospital mortality, patients receiving adjuvant chemotherapy, intervals between surgery to chemotherapy, and chemotherapy cycles did not differ between G3 and G4.

### 3.4. Predictors of Disease-Free and Overall Survival in G3 and G4 Patients

Significant prognostic factors of DFS in univariate analysis in G3 and G4 patients included nodal status (*p* < 0.001), stage (*p* < 0.0001), LNR (*p* < 0.0001), lymphatic invasion (*p* = 0.002), perineural invasion (*p* = 0.023), and adjuvant chemotherapy (*p* = 0.046), as shown in Table 5. After adjusting for confounder in multivariate analysis, stage and administration of adjuvant chemotherapy were independent factors affecting recurrence (Table 5). In univariate analysis, nodal status (*p* < 0.001), stage (*p* < 0.0001), LNR (*p* < 0.0001), and the presence of lymphatic (*p* = 0.003) or perineural invasion (*p* = 0.006) were the factors affecting OS (Table 6). Tumor stage was the only independent prognostic factor for OS in multivariate analysis. Compared with stage II disease, the HRs were 7.074 (95% CI, 1.532–32.667; *p* = 0.012), 8.942 (95% CI, 1.712–46.704; *p* = 0.009), and 12.450 (95% CI, 2.265–68.439; *p* = 0.004) for stages IIIA, IIIB and IIIC, respectively (Table 6). There was no difference in median DFS (12.0 vs. 11.4 months, *p* = 0.809) and OS (18.8 vs. 17.8 months, *p* = 0.964) between G3 and G4 patients (Figure 2).

### 3.5. Recurrence Rates and Patterns

Table 7 shows a comparison of the recurrence rates and patterns of recurrence between G1 and G2 patients as well as between G3 and G4 patients. G1 patients had higher recurrence rates than G2 patients (*p* = 0.026). The recurrence rate did not differ between G3 and G4 patients (*p* = 0.408). Similar recurrence patterns were found in G1 and G2. G4 patients had greater rates of local/regional recurrence than G3 patients (*p* = 0.010). In addition, a statistically significant difference in recurrence pattern was found between G3 and G4 patients (*p* = 0.008).

## 4. Discussion

Long-term outcomes in patients with locally advanced resectable GC still remain unsatisfactory despite aggressive extensive surgery followed by adjuvant chemotherapy [3,4,5,6,7,8,9,10,11]. The impact of PR on survival in patients with T4b GC is unknown. Few studies in the literature addressed the issue whether further Whipple’s operation with clear distal duodenal margins can improve GC patient survival or not. To the best of our knowledge, our study is the largest cohort study aimed at evaluating the impact of PR, including Whipple’s operation and distal pancreatectomy, on survival in patients with locally advanced resectable GC. Our present study indicated that PR was an independent unfavorable prognostic factor for DFS and OS in patients with T4b lesion. G4 patients had higher rates of complications and loco-regional recurrence than G3 patients; furthermore, Whipple’s operation aiming to achieve R0 resection did not prolong DFS or OS.

Studies showed that median OS was 25.9–27 months in T4b patients undergoing surgical resection [3,4]. Our previous research indicated that locally advanced GC patients undergoing MOR had 31.6 months of mean survival time (range, 21.9–41.2). In addition, we also found that involvement of the liver was associated with better survival than other organ invasion in multivariate analysis (HR = 4.49, 95% CI, 1.89–10.67; *p* = 0.0001). Our present results showed that there was no significant difference in demographics except perineural invasion (higher in G2) between G1 and G2 patients. In multivariate analysis of prognostic factors for DFS and OS, PR independently affected T4b patient prognosis (Table 2 and Table 3), and G1 patients had better DFS and OS curves than G2 (Figure 1) suggesting that GC with pancreatic invasion appeared to have different/aggressive tumor behavior as compared with that with other adjacent organ involvement.

Theoretically, patients with positive duodenal margins should benefit from undergoing an additional Whipple’s operation to achieve R0 resection. Our previous results have shown that patients who underwent MOR and R0 resection had survival benefit compared with those with MOR and R1/R2 resection (15.7 vs. 11.2 months; *p* = 0.007) [5]. However, the OS did not differ between pathological T4b patients who underwent MOR and clinical T4b patients who did not undergo MOR (13.9 vs. 11.2 months; *p* = 0.457) [5]. The current study aimed to assess the impact of Whipple’s operation to achieve R0 resection on survival in locally advanced GC patients. Our data (Figure 2) revealed that there was no difference in median DFS (11.4 vs. 12.0 months) or OS (17.8 vs. 18.8 months) between the patients undergoing additional Whipple’s operation (G4) and those with positive duodenal margins not proceeding with further Whipple’s operation (G3). Compared with G3, G4 had higher complication rates (45.7% vs. 25.5%; *p* = 0.016). Similar to our findings, a nationwide study from the Dutch upper gastrointestinal cancer audit group that evaluated the outcomes of gastrectomies with PR in patients with GC showed that Clavien–Dindo grade ≥ III complications occurred in 21 of 55 patients (38%), which was significantly higher than in those who did not have additional PR (*p* < 0.001); further, the median OS was only 15 months in patients who underwent additional PR [11].

Several studies have indicated that surgical complications adversely affected long-term DSF and OS in GC patients after radical gastrectomy, which was more significant in those with major and/or infectious complications [14,15,16,17,18]. Our results showed higher rates of complication and local/regional recurrence in G4 patients than in G3 patients, suggesting that more extensive surgery involving Whipple’s operation did not improve and/or enhance local control that may be partially explained by greater percentages of complication in the G4 group. Surgical complications, mainly infections, induce excessive production of pro-inflammatory and inflammatory cytokines, such as tumor necrosis factor -alpha and interleukins 1 and 6, which may result in immune suppression promoting cancer growth and metastasis [19,20,21]. Nonetheless, considering other confounding prognostic factors, complications did not influence the DFS and OS in G3 and G4 group patients, and the stage alone was an independent factor influencing patient OS. Our data implied that disease biological behavior and stage were the most important determinants affecting outcomes in patients with invasion of the pancreatic head.

There is a lack of global consensus regarding whether locally advanced resectable GC should be treated with up-front surgery, or there is a role of perioperative chemotherapy or preoperative chemoradiation. In Asia, patients usually undergo radical resection including MOR followed by adjuvant chemotherapy [22,23]. In Europe, perioperative chemotherapy is recommended to GC patients based on the results of MAGIC or FLOT4 trial [24,25]. However, in North America surgical resection followed by chemoradiotherapy is preferred based on the Intergroup 0116 studies [26]. Our present findings revealed that patients undergoing PR had unfavorable prognosis and Whipple’s operation did not prolong their survival. Therefore, other treatment strategies should be considered to improve surgical results in locally advanced resectable GC patients with clinical pancreatic invasion. The rationale for neoadjuvant chemotherapy approach is that there are several advantages of administration of chemotherapy before surgery. First, the response to chemotherapeutic regimen can be evaluated or confirmed, which can be either guided as adjuvant chemotherapy or prevent fruitless extensive surgery. Second, downstaging the tumor may lead to higher R0 resection rates and avoid Whipple’s operation. Third, occult metastasis caused by tumor cell dissemination can be treated at an earlier point when the patient is usually in a better general health condition. In this regard, a recent phase III randomized study (PRODIGY) has indicated that neoadjuvant chemotherapy followed by surgery had favorable results as compared to up-front surgery in resectable advanced GC as evidenced by higher R0 resection rates (96.4% vs. 85.8%; *p* < 0.0001), lower pathological stage with pathological complete response (10.4% vs. 0%; *p* < 0.0001) and greater 3-year DFS rates (66.3% vs. 60.2%; HR = 0.70; 95% CI, 0.52–0.95; *p* = 0.023) [27]. Furthermore, another study also demonstrated that perioperative chemotherapy was associated with longer median OS than up-front surgery did [11]. Nonetheless, further randomized controlled trials are required to confirm whether GC patients with pancreatic invasion will benefit from neoadjuvant therapy followed by surgery as compared with up-front surgery.

Other novel treatment strategies have been proposed to improve patient survival. The ATTRACTION-04 trial and KEYNOTE-059 have revealed that the combination of chemotherapy with immune checkpoint inhibitor in locally advanced GC might be considered to enhance patient outcomes [28,29]. The exploratory analysis of MAGIC trial showed that patients with microsatellite instability high or deficient mismatch repair GC might be responsive to immune checkpoint inhibitor therapy but are resistant to conventional chemotherapy [30]. The addition of anti-HER-2 antibodies to conventional chemotherapy and/or immune checkpoint inhibitor in HER-2 positive resectable GC [31] are under investigation and the trial results will be reported in the near future.

Our study had several limitations. First, owing to retrospective nature of the study design, possible selection bias exists, including the extent of radical surgery which could have been affected by the surgeon’s preference, and also the existing comorbidities and age of the patients. Second, the time span of this study was as long as 20 years; therefore, the strategies of pT4b management, additionally one or multiple organ invasion, and positive duodenal margin will differ over time. Third, the recurrent patients received varied salvage chemotherapy regimens that may have influenced the survival time. Nonetheless, our results are likely to demonstrate that PR resulted in poorer survival in patients with T4b lesion, and Whipple’s operation that aimed to achieve clear duodenal margins did not prolong survival.

## 5. Conclusions

In conclusion, our results suggest that GC with pancreatic invasion indicates not only anatomic involvement but also more aggressive tumor biologic behavior. The surgical prognosis in patients with pancreatic involvement was significantly poorer than in those with other adjacent organs invasion. Additional Whipple’s operation to achieve R0 resection did not improve survival as compared to gastrectomy with positive duodenal margins. Other strategies such as neoadjuvant chemotherapy followed by surgery might be considered in treating patients who did not have evidence of tumor bleeding but have clinical pancreatic invasion to improve their outcomes.

## Figures and Tables

**Figure 1 cancers-13-01289-f001:**
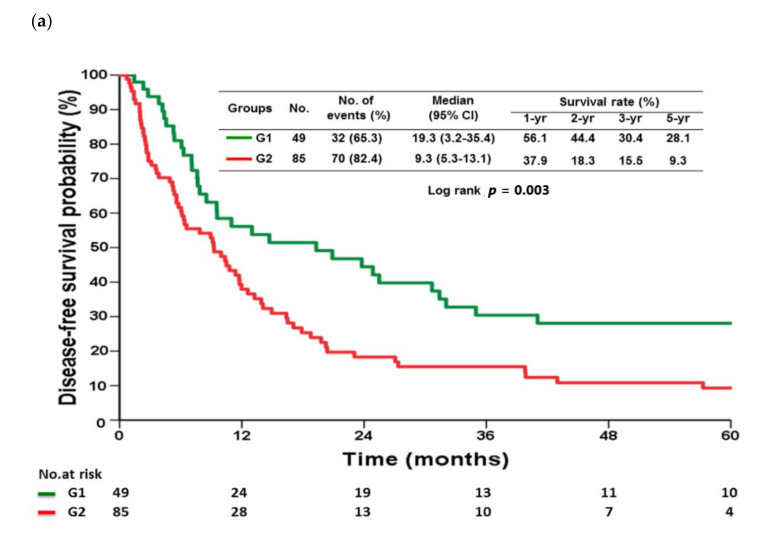
Disease-free survival (**a**) and overall survival (**b**) between G1 and G2 patients.

**Figure 2 cancers-13-01289-f002:**
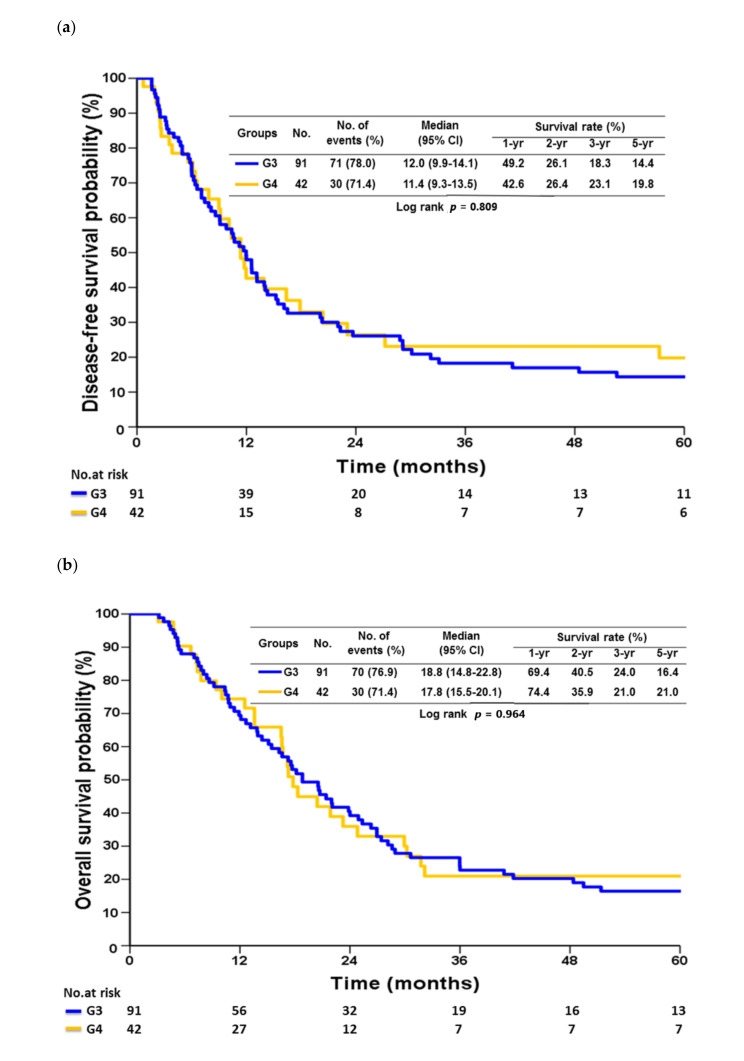
Disease-free survival (**a**) and overall survival (**b**) between G3 and G4 patients.

**Table 1 cancers-13-01289-t001:** Demographics and clinicopathological features between group 1 and 2 patients.

Variables	Group 1	Group 2	*p* Value
No. of patients	50	94	
Age (years), mean ± SD	62.4 ± 14.0	63.2 ± 12.2	0.731
Gender			0.722
Male	36 (72.0)	65 (69.1)	
Female	14 (28.0)	29 (30.9)	
Tumor size (cm), mean ± SD	7.4 ± 2.8	7.1 ± 3.3	0.599
Tumor location			0.074
Upper	12 (24.0)	33 (35.1)	
Middle	9 (18.0)	5 (5.3)	
Lower	25 (50.0)	46 (48.9)	
Whole	4 (8.0)	10 (10.6)	
Type of gastrectomy			0.521
Total	32 (64.0)	55 (58.5)	
Subtotal	18 (36.0)	39 (41.5)	
Nodal status			0.129
N0	12 (24.0)	10 (10.6)	
N1	4 (8.0)	7 (7.4)	
N2	10 (20.0)	23 (24.5)	
N3a	9 (18.0)	31 (33.0)	
N3b	15 (30.0)	23 (24.5)	
Stage			0.105
IIIA	12 (24.0)	10 (10.6)	
IIIB	14 (28.0)	30 (31.9)	
IIIC	24 (48.0)	54 (57.5)	
No. of lymph node retrieval, mean ± SD	36.8 ± 20.3	37.7 ± 18.0	0.777
LNR, mean ± SD	0.32 ± 0.31	0.32 ± 0.27	0.968
Differentiation			0.387
Yes	13 (26.0)	31 (33.0)	
No	37 (74.0)	63 (67.0)	
Lymphatic invasion			0.290
Yes	36 (72.0)	75 (79.8)	
No	14 (28.0)	19 (20.2)	
Vascular invasion			0.387
Yes	14 (28.0)	33 (35.1)	
No	36 (72.0)	61 (64.9)	
Perineural invasion			<0.001
Yes	25 (50.0)	75 (79.8)	
No	25 (50.0)	19 (20.2)	
Complication	11 (22.0)	35 (37.2)	0.062
Hospital mortality	1 (2.0)	9(9.6)	0.165
Adjuvant chemotherapy	36 (72.0)	61 (64.9)	0.457
Intervals between surgery to chemotherapy (months), mean ± SD	1.9 ± 1.3	1.7 ± 1.1	0.304
Chemotherapy cycles, mean ± SD	8.8 ± 8.4	7.2 ± 5.3	0.287

Group 1: pT4b without pancreatic resection; Group 2: pT4b with pancreatic resection. LNR, metastatic to retrieved lymph node ratio; SD, standard deviation.Values in parentheses are percentages.

**Table 2 cancers-13-01289-t002:** Univariate and multivariate analysis of prognostic factors for disease-free survival in group 1 and 2 patients.

Factors	Median (Months)	95% CI	*p* Value	Hazard Ratios	95% CI	*p* Value
Age			0.668			
≤65 (*n* = 74)	12.0	6.2–17.8				
>65 (*n* = 60)	9.6	6.5–12.6				
Gender			0.057			
Male (*n* = 94)	13.0	6.8–19.1				
Female (*n* = 40)	7.9	4.3–11.4				
Tumor size (cm)			0.458			
≤6.5 (*n* = 71)	11.7	8.0–15.5				
>6.5 (*n* = 63)	9.2	5.5–13.0				
Location			0.903			
Upper (*n* = 40)	10.0	3.9–16.0				
Middle (*n* = 13)	9.5	0.1–19.7				
Lower (*n* = 68)	11.4	7.1–15.6				
Whole (*n* = 13)	9.2	5.0–13.5				
Type of gastrectomy			0.206			
Total (*n* = 78)	9.2	6.6–11.9				
Subtotal (*n* = 56)	12.0	7.7–16.2				
Pancreatic resection			0.003			
No, Group 1 (*n* = 49)	19.3	3.2–35.4		1		
Yes, Group 2 (*n* = 85)	9.3	5.3–13.2		1.740	1.116–2.713	0.015
Nodal status			<0.0001			
N0 (*n* = 20)	41.0	9.7–72.4				
N1 (*n* = 10)	7.9	3.0–12.7				
N2 (*n* = 30)	14.9	6.6–23.3				
N3a (*n* = 39)	12.6	4.3–20.9				
N3b (*n* = 35)	6.1	5.2–77.0				
Stage			<0.001			
IIIA (*n* = 20)	41.0	9.7–72.4		1		
IIIB (*n* = 40)	10.8	7.6–14.0		1.542	0.363–6.551	0.558
IIIC (*n* = 74)	7.1	4.0–10.2		1.803	0.330–9.854	0.497
LNR			<0.0001			
≤0.04 (*n* = 25)	41.0	12.8–69.3		1		
>0.04, ≤0.41 (*n* = 67)	10.8	7.3–14.3		1.546	0.440–5.432	0.497
>0.41 (*n* = 42)	6.3	5.2–7.3		3.109	0.792–12.198	0.104
Differentiation			0.352			
No (*n* = 42)	11.7	6.2–17.2				
Yes (*n* = 92)	10.4	7.8–13.0				
Lymphatic invasion			0.020			
No (*n* = 30)	20.9	14.3–27.5		1		
Yes (*n* = 104)	9.3	6.3–12.3		0.743	0.359–1.538	0.424
Vascular invasion			0.117			
No (*n* = 91)	11.4	7.1–15.6				
Yes (*n* = 43)	7.9	3.0–12.8				
Perineural invasion			0.003			
No (*n* = 42)	13.3	2.3–24.2		1		
Yes (*n* = 92)	9.3	6.2–12.3		1.256	0.772–2.043	0.359
Complication			0.850			
No (*n* = 95)	11.0	8.1–13.9				
Yes (*n* = 39)	9.2	4.6–13.9				
Adjuvant chemotherapy			0.160			
No (*n* = 41)	8.5	5.0–12.0				
Yes (*n* = 93)	11.4	8.3–14.5				

Group 1: pT4b without pancreatic resection; Group 2: pT4b with pancreatic resection. CI, confidence interval; LNR, ratio of metastatic to retrieved lymph nodes.

**Table 3 cancers-13-01289-t003:** Univariate and multivariate analysis of prognostic factors for overall survival in group 1 and 2 patients.

Factors	Median (Months)	95% CI	*p* Value	Hazard Ratios	95% CI	*p* Value
Age			0.247			
≤65 (*n* = 74)	22.7	15.1–30.2				
>65 (*n* = 60)	16.6	12.9–20.2				
Gender			0.257			
Male (*n* = 94)	20.4	15.1–25.7				
Female (*n* = 40)	16.0	9.9–22.2				
Tumor size (cm)			0.529			
≤6.5 (*n* = 71)	17.8	10.4–25.2				
>6.5 (*n* = 63)	18.0	15.2–20.8				
Location			0.774			
Upper (*n* = 40)	13.5	6.7–20.2				
Middle (*n* = 13)	15.8	0.1–33.2				
Lower (*n* = 68)	18.3	12.6–24.0				
Whole (*n* = 13)	18.4	14.0–22.7				
Type of gastrectomy			0.123			
Total (*n* = 78)	17.4	12.4–22.3				
Subtotal (*n* = 56)	21.9	15.2–28.6				
Pancreatic resection			0.004			
No, Group 1 (*n* = 49)	27.1	14.2–40.1		1		
Yes, Group 2 (*n* = 85)	16.0	12.1–20.0		1.897	1.210–2.974	0.005
Nodal status			<0.0001			
N0 (*n* = 20)	61.1	0.1–143.3				
N1 (*n* = 10)	12.6	8.8–16.4				
N2 (*n* = 30)	23.1	13.3–32.9				
N3a (*n* = 39)	16.6	10.8–22.4				
N3b (*n* = 35)	12.7	10.5–15.0				
Stage			<0.001			
IIIA (*n* = 20)	61.1	0.1–143.3		1		
IIIB (*n* = 40)	22.7	15.3–30.1		1.225	0.296–5.072	0.779
IIIC (*n* = 74)	13.6	10.8–16.4		1.368	0.262–7.140	0.710
LNR			<0.0001			
≤0.04 (*n* = 25)	51.8	0.1–107.6		1		
>0.04, ≤0.41 (*n* = 67)	17.8	11.6–24.0		1.622	0.464–5.665	0.449
>0.41 (*n* = 42)	12.7	10.7–14.8		3.720	0.957–14.462	0.058
Differentiation			0.053			
No (*n* = 42)	21.4	16.1–26.7				
Yes (*n* = 92)	15.8	10.7–20.8				
Lymphatic invasion			0.009			
No (*n* = 30)	28.4	19.8–37.1		1		
Yes (*n* = 104)	15.8	12.4–19.2		0.931	0.455–1.902	0.844
Vascular invasion			0.053			
No (*n* = 91)	21.4	16.1–26.7				
Yes (*n* = 43)	15.8	10.7–20.8				
Perineural invasion			0.004			
No (*n* = 42)	23.2	14.3–2.1		1		
Yes (*n* = 92)	17.2	13.1–21.3		1.262	0.774–2.056	0.351
Complication			0.981			
No (*n* = 95)	17.7	11.9–23.5				
Yes (*n* = 39)	18.3	10.36–26.3				
Adjuvant chemotherapy			0.111			
No (*n* = 41)	13.1	6.6–19.5				
Yes (*n* = 93)	20.4	15.2–25.7				

Group 1: pT4b without pancreatic resection; Group 2: pT4b with pancreatic resection. CI, confidence interval; LNR, ratio of metastatic to retrieved lymph nodes.

**Table 4 cancers-13-01289-t004:** Demographics and clinicopathological features between group 3 and 4 patients.

Variables	Group 3	Group 4	*p* Value
No. of patients	98	46	
Age (years), mean ± SD	63.7 ± 12.6	61.7 ± 13.3	0.381
Gender			0.523
Male	65 (66.3)	28 (60.9)	
Female	33 (33.7)	18 (39.1)	
Tumor size (cm), mean ± SD	6.5 ± 3.0	6.6 ± 2.8	0.834
Tumor location			0.189
Upper	7 (7.1)	1 (2.2)	
Middle	6 (6.1)	0	
Lower	78 (79.6)	43 (93.5)	
Whole	7 (7.1)	2 (4.3)	
Type of gastrectomy			0.535
Total	25 (25.5)	14 (30.4)	
Subtotal	73 (74.5)	32 (69.6)	
T status			<0.0001
T3	7 (7.1)	1 (2.2)	
T4a	79 (80.7)	13 (28.3)	
T4b	12 (12.2)	32 (69.6)	
Nodal status			0.017
N0	8 (8.2)	8 (17.4)	
N1	6 (6.1)	6 (13.0)	
N2	14 (14.3)	9 (19.6)	
N3a	30 (30.6)	16 (34.8)	
N3b	40 (40.8)	7 (15.2)	
Stage			0.940
IIB	7 (7.1)	2 (4.3)	
IIIA	19 (19.4)	10 (21.7)	
IIIB	28 (28.6)	14 (30.4)	
IIIC	44 (44.9)	20 (43.5)	
No. of lymph node retrieval, mean ± SD	31.5 ± 13.8	38.3 ± 16.0	0.009
LNR, mean ± SD	0.48 ± 0.32	0.24 ± 0.22	<0.0001
Differentiation			0.574
Yes	21 (21.4)	8 (25.0)	
No	77 (78.6)	24 (75.0)	
Lymphatic invasion			0.052
Yes	82 (83.7)	32 (69.6)	
No	16 (16.3)	14 (30.4)	
Vascular invasion			0.969
Yes	28 (28.6)	13 (28.3)	
No	70 (71.4)	33 (71.7)	
Perineural invasion			0.536
Yes	75 (76.5)	33 (71.7)	
No	23 (23.5)	13 (28.3)	
Complication	25 (25.5)	21 (45.7)	0.016
Hospital mortality	7 (7.1)	4 (8.7)	0.744
Adjuvant chemotherapy	63 (64.3)	30 (65.2)	0.913
Intervals between surgery to chemotherapy (months), mean ± SD	1.5 ± 1.0	1.8 ± 0.8	0.132
Chemotherapy cycles, mean ± SD	7.9 ± 9.5	8.5 ± 5.9	0.792

Group 3: distal margin (+) without Whipple’s operation; Group 4: cT4b with Whipple’s operation. LNR, metastatic to retrieved lymph node ratio; SD, standard deviation. Values in parentheses are percentages.

**Table 5 cancers-13-01289-t005:** Univariate and multivariate analysis of prognostic factors for disease-free survival in group 3 and 4 patients.

Factors	Median (Months)	95% CI	*p* Value	Hazard Ratios	95% CI	*p* Value
Age			0.953			
≤65 (*n* = 67)	12.0	8.1–15.8				
>65 (*n* = 66)	11.4	9.1–13.6				
Gender			0.678			
Male (*n* = 86)	11.7	9.2–14.2				
Female (*n* = 47)	11.4	8.6–14.2				
Tumor size (cm)			0.522			
≤6.0 (*n* = 72)	11.7	9.1–14.3				
>6.0 (*n* = 61)	11.4	8.2–14.6				
Location			0.711			
Upper (*n* = 7)	12.6	0.1–28.9				
Middle (*n* = 6)	12.6	0.1–25.1				
Lower (*n* = 113)	11.4	9.5–13.2				
Whole (*n* = 7)	15.5	10.0–20.9				
Type of gastrectomy			0.150			
Total (*n* = 34)	10.4	5.4–15.4				
Subtotal (*n* = 99)	12.0	9.5–14.4				
Duodenal margins			0.809			
Positive, Group 3 (*n* = 91)	12.0	9.9–14.1				
Negative, Group 4 (*n* = 42)	11.4	9.3–13.5				
T status			0.074			
T3 (*n* = 8)	11.2	6.8–15.7				
T4a (*n* = 86)	12.6	10.2–14.9				
T4b (*n* = 39)	7.9	4.4–11.4				
Nodal status						
N0 (*n* = 15)	NA		<0.001			
N1 (*n* = 11)	11.8	6.3–17.3				
N2 (*n* = 22)	11.7	8.4–15.0				
N3a (*n* = 44)	12.0	9.3–14.6				
N3b (*n* = 41)	7.1	3.3–10.9				
Stage						
II (*n* = 9)	NA		<0.0001	1		
IIIA (*n* = 28)	14.3	9.8–18.8		7.177	1.569–32.828	0.011
IIIB (*n* = 38)	12.0	10.2–13.8		12.507	2.439–64.141	0.002
IIIC (*n* = 58)	7.4	3.8–11.0		18.754	3.467–101.458	0.001
LNR			<0.0001			
≤0.10 (*n* = 28)	22.3	8.0–36.5		1		
>0.10 (*n* = 105)	10.4	7.6–13.1		1.070	0.506–2.266	0.859
Differentiation			0.879			
No (*n* = 98)	12.0	9.7–14.3				
Yes (*n* = 35)	10.7	7.9–13.4				
Lymphatic invasion			0.002			
No (*n* = 28)	10.7	8.3–13.1		1		
Yes (*n* = 105)	22.3	10.6–34.0		0.778	0.415–1.458	0.433
Vascular invasion			0.112	-		
No (*n* = 95)	12.0	10.4–13.6				
Yes (*n* = 38)	7.9	4.2–11.5				
Perineural invasion			0.023			
No (*n* = 34)	12.0	10.1–13.9		1		
Yes (*n* = 99)	11.4	8.0–14.8		1.068	0.635–1.796	0.803
Complication			0.997			
No (*n* = 97)	11.4	9.9–12.8				
Yes (*n* = 36)	12.6	7.7–17.4				
Adjuvant chemotherapy			0.046			
No (*n* = 40)	9.1	4.6–13.7		1.904	1.188–3.053	0.007
Yes (*n* = 93)	12.6	9.9–15.3		1		

Group 3: distal margin (+) without Whipple’s operation; Group 4: cT4b with Whipple’s operation. CI, confidence interval; LNR, ratio of metastatic to retrieved lymph nodes; NA, not available.

**Table 6 cancers-13-01289-t006:** Univariate and multivariate analysis of prognostic factors for overall survival in group 3 and 4 patients.

Factors	Median (Months)	95% CI	*p* Value	Hazard Ratios	95% CI	*p* Value
Age			0.824			
≤65 (*n* = 67)	21.9	14.8–28.9				
>65 (*n* = 66)	16.6	12.8–20.3				
Gender			0.781			
Male (*n* = 86)	20.4	15.7–25.2				
Female (*n* = 47)	18.2	16.5–19.8				
Tumor size (cm)			0.617			
≤6.0 (*n* = 72)	17.8	14.0–21.6				
>6.0 (*n* = 61)	18.8	14.1–23.5				
Location			0.669			
Upper (*n* = 7)	27.4	6.1–48.8				
Middle (*n* = 6)	18.8	16.4–21.2				
Lower (*n* = 113)	18.2	14.5–21.8				
Whole (*n* = 7)	20.4	9.0–31.8				
Type of gastrectomy			0.084			
Total (*n* = 34)	13.9	7.3–20.6				
Subtotal (*n* = 99)	20.6	16.2–24.9				
Duodenal margins			0.964			
Positive, Group 3 (*n* = 91)	18.8	14.8–22.8				
Negative, Group 4 (*n* = 42)	17.8	15.5–20.1				
T status			0.109			
T3 (*n* = 8)	20.6	9.3–31.8				
T4a (*n* = 86)	20.7	14.7–26.7				
T4b (*n* = 39)	16.6	10.4–22.8				
Nodal status			<0.001			
N0 (*n* = 15)	-					
N1 (*n* = 11)	15.1	0.6–29.6				
N2 (*n* = 22)	17.2	9.9–24.5				
N3a (*n* = 44)	17.8	13.3–22.3				
N3b (*n* = 41)	13.9	6.1–21.8				
Stage			<0.0001			
II (*n* = 9)	NA			1		
IIIA (*n* = 28)	20.4	15.5–25.3		7.074	1.532–32.667	0.012
IIIB (*n* = 38)	18.2	12.4–23.9		8.942	1.712–46.704	0.009
IIIC (*n* = 58)	13.9	8.1–19.7		12.450	2.265–68.439	0.004
LNR			<0.0001			
≤0.10 (*n* = 28)	26.3	16.3–36.3		1		
>0.10 (*n* = 105)	17.2	14.6–19.8		1.409	0.673–2.950	0.363
Differentiation			0.358			
No (*n* = 98)	18.2	15.0–21.3				
Yes (*n* = 35)	22.1	11.3–32.8				
Lymphatic invasion			0.003			
No (*n* = 28)	26.3	16.3–36.3		1		
Yes (*n* = 105)	17.2	15.7–18.7		0.723	0.383–1.364	0.316
Vascular invasion			0.187			
No (*n* = 95)	18.8	15.7–21.9				
Yes (*n* = 38)	15.1	9.1–21.2				
Perineural invasion			0.006			
No (*n* = 34)	22.1	14.39–29.2		1		
Yes (*n* = 99)	17.3	12.6–21.9		1.206	0.724–2.007	0.472
Complication			0.656			
No (*n* = 97)	18.8	15.5–22.2				
Yes (*n* = 36)	18.3	13.9–22.7				
Adjuvant chemotherapy			0.060			
No (*n* = 40)	12.7	0.1–25.9				
Yes (*n* = 93)	20.6	16.2–25.0				

Group 3: distal margin (+) without Whipple’s operation; Group 4: cT4b with Whipple’s operation. CI, confidence interval; LNR, ratio of metastatic to retrieved lymph nodes, NA, nor available.

**Table 7 cancers-13-01289-t007:** Recurrence rate and pattern.

Variables	Group 1 (*n* = 49)	Group 2 (*n* = 85)	*p* Value	Group 3 (*n* = 91)	Group 4 (*n* = 42)	*p* Value
Recurrence			0.026			0.408
Yes	32 (65.3)	70 (82.4)		71 (78.0)	30 (71.4)	
No	17 (34.7)	15 (17.6)		20 (22.0)	12 (28.6)	
Recurrence pattern						
Local/regional (L)	13 (26.5)	32 (37.6)	0.189	23 (25.3)	20 (47.6)	0.010
Hematogenous (H)	18 (36.7)	30 (35.3)	0.867	28 (30.8)	10 (23.8)	0.409
Peritoneal (P)	13 (26.5)	26 (30.6)	0.618	35 (38.5)	16 (38.1)	0.968
Recurrence pattern			0.122			0.008
None	17 (34.7)	15 (17.6)		20 (22.0)	12 (28.6)	
L	4 (8.2)	17 (20.0)		13 (14.3)	7 (16.7)	
L + H	7 (14.3)	7 (8.2)		5 (5.5)	5 (11.9)	
H	8 (16.3)	20 (23.5)		18 (19.8)	2 (4.8)	
H + P	3 (6.1)	2 (2.4)		4 (4.4)	3 (7.1)	
L + P	2 (4.1)	7 (8.2)		4 (4.4)	8 (19.0)	
P	8 (16.3)	16 (18.8)		26 (28.6)	5 (11.9)	
L + H + P	0	1 (1.2)		1 (1.0)	0	

Group 1: pT4b without pancreatic resection; Group 2: pT4b with pancreatic resection; Group 3: distal margin (+) without Whipple’s operation; Group 4: cT4b with Whipple’s operation. Values in parentheses are percentages.

## Data Availability

No new data were created or analyzed in this study. Data sharing is not applicable to this article.

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
