# Peer review of "Impact of Pancreatic Resection on Survival in Locally Advanced Resectable Gastric Cancer"

_cancers, 2021, doi:10.3390/cancers13061289_

Round 1

Reviewer 1 Report

In this article, entitled "Impact of pancreatic resection on survival in locally advanced 2 resectable gastric cancer" the Authors tried to answer a very important question whether Gastric adenocarcinoma (GC)  patients are benefitted from pancreatic resection (PR). This evaluation is important for making the appropriate decision regarding surgery and can have a great impact on patient life.

While this is a nicely designed study and answered many questions, I have some concerns as indicated below:

1.In the introduction section impact of pancreatic resection (PR) in GC patients should discuss more, than only GC. It is important to understand the study question that when and why PR is necessary for GC patients and when it can be avoided.  

  1. Table 1 and its description should be in the method section not in the results.

3.Results should be more organized in three to four sub-headings,  indicating each observation. Under each subheading, detailed results should be discussed.

  1. Figure legends should be discussed in detail. Tables within the graph should be clearly labeled and described in the text.

Author Response

In this article, entitled "Impact of pancreatic resection on survival in locally advanced resectable gastric cancer" the Authors tried to answer a very important question whether Gastric adenocarcinoma (GC) patients are benefited from pancreatic resection (PR). This evaluation is important for making the appropriate decision regarding surgery and can have a great impact on patient life. While this is a nicely designed study and answered many questions, I have some concerns as indicated below:

Response: We thank the reviewer very much for this important remark that our manuscript is a nicely designed study addressing the issue of whether gastric adenocarcinoma (GC) patients are benefitted from pancreatic resection (PR).

Question 1: In the introduction section impact of pancreatic resection (PR) in GC patients should discuss more, than only GC. It is important to understand the study question that when and why PR is necessary for GC patients and when it can be avoided.

Response: We thank greatly the reviewer’s comments and have added some sentences in the introduction section to emphasize the impact of additional pancreatic resection on GC patient outcomes.

Question 2: Table 1 and its description should be in the method section not in the results.

Response: We thank for the reviewer’s suggestion. We respectfully submit that we think Table 1 is better at the section of “Results” since Table 4 is at the section of “Results”.

Question 3: Results should be more organized in three to four sub-headings, indicating each observation. Under each subheading, detailed results should be discussed.

Response: We have added subheadings in the “Results” and results also have been described/discussed properly in the revised manuscript.

Question 4: Figure legends should be discussed in detail. Tables within the graph should be clearly labeled and described in the text.

Response: We have made some corrections according to reviewer’s comments in the revised article. Please refer to the part of revised manuscript, highlighted in yellow. The results of Figure 2 have been well discussed in the paragraph 4 of “Discussion”.

Reviewer 2 Report

Dear Authors,

I would like to congratulate on such splendid paper which required a lot of work (a very long study) which is extremely impressive! The paper is written very well so thus it is organized properly as well. I have several minor suggestions:

  • Line 87 - data has been obtained from the database, however, it would be beneficial to add an information who performed the histopathological examination - it would be beneficial to add that it was performed by the histopathologist who specializes in this field and also add the name of the institution from which he or she comes.
  • I would indicate on the exclusion and inclusion criteria of patients.
  • Line 206 a word 'Discussion' should be highlighted
  • Please check the manuscript once again in terms of English since I have detected several (however very small) mistakes or grammatical erros

I would like to wish you all the best with your further research!

Best regards

Reviewer

Author Response

Reviewer 2:

I would like to congratulate on such splendid paper which required a lot of work (a very long study) which is extremely impressive! The paper is written very well so thus it is organized properly as well. I have several minor suggestions:

Response: We gratefully thank for the reviewer’s comments that our manuscript is well written and is organized properly.

Question 1: Line 87 - data has been obtained from the database, however, it would be beneficial to add information that performed the histopathological examination - it would be beneficial to add that it was performed by the histopathologist who specializes in this field and also add the name of the institution from which he or she comes.

Response: We have added the senior experienced histopathologist in this filed in the authorship.

Question 2: I would indicate on the exclusion and inclusion criteria of patients.

Response: We have added subheadings indicating exclusion and inclusion criteria in the method section.

Question 3: Line 206 a word 'Discussion' should be highlighted

Response: We have made a change in the revised manuscript.

Question 4: Please check the manuscript once again in terms of English since I have detected several (however very small) mistakes or grammatical errors.

Response: This article has been modified by native English speaker. We also have gone through the manuscript carefully and corrected for the errors.